# Predicting the effectiveness of the online clinical clerkship curriculum: Development of a multivariate prediction model and validation study

**Naoto Kuroda**[1,2]*, **Anna Suzuki**[3], **Kai Ozawa**[4], **Nobuhiro Nagai**[5], **Yurika Okuyama**[6], **Kana Koshiishi**[7], **Masafumi Yamada**[5], **Makoto Kikukawa**[8]

1 Department of Pediatrics, Wayne State University, Detroit, Michigan, United States of America,
2 Department of Epileptology, Tohoku University Graduate School of Medicine, Sendai, Japan, 3 Kyushu University School of Medicine, Fukuoka, Japan, 4 The Jikei University School of Medicine, Tokyo, Japan,
5 Faculty of Medicine, Shimane University, Izumo, Japan, 6 Akita University School of Medicine, Akita, Japan, 7 Asahikawa Medical University, Asahikawa, Japan, 8 Department of Medical Education, Kyushu University, Fukuoka, Japan

* naoto.kuroda@wayne.edu

**Data Availability Statement:** All relevant data are within the paper and its Supporting Information files.

## Abstract

Given scientific and technological advancements, expectations of online medical education are increasing. However, there is no way to predict the effectiveness of online clinical clerkship curricula. To develop a prediction model, we conducted cross-sectional national surveys in Japan. Social media surveys were conducted among medical students in Japan during the periods May–June 2020 and February–March 2021. We used the former for the derivation dataset and the latter for the validation dataset. We asked students questions in three areas: 1) opportunities to learn from each educational approach (lectures, medical quizzes, assignments, oral presentations, observation of physicians' practice, clinical skills practice, participation in interprofessional meetings, and interactive discussions with physicians) in online clinical clerkships compared to face-to-face, 2) frequency of technical problems on online platforms, and 3) satisfaction and motivation as outcome measurements. We developed a scoring system based on a multivariate prediction model for satisfaction and motivation in a cross-sectional study of 1,671 medical students during the period May–June 2020. We externally validated this scoring with a cross-sectional study of 106 medical students during February–March 2021 and assessed its predictive performance. The final prediction models in the derivation dataset included eight variables (frequency of lectures, medical quizzes, oral presentations, observation of physicians' practice, clinical skills practice, participation in interprofessional meetings, interactive discussions with physicians, and technical problems). We applied the prediction models created using the derivation dataset to a validation dataset. The prediction performance values, based on the area under the receiver operating characteristic curve, were 0.69 for satisfaction (sensitivity, 0.50; specificity, 0.89) and 0.75 for motivation (sensitivity, 0.71; specificity, 0.85). We developed a prediction model for the effectiveness of the online clinical clerkship curriculum, based on

**Funding:** M.K. was supported by JSPS KAKENHI (grant number 17H04097). URL: https://www.jsps.go.jp/english/index.html This funder did not any role in this study design, data collection and analysis, decision to publish, or preparation of the manuscript.

**Competing interests:** The authors have declared that no competing interests exist.

students' satisfaction and motivation. Our model will accurately predict and improve the online clinical clerkship curriculum effectiveness.

## Introduction

Given scientific and technological advancements, the medical education community's expectations of online education are rising [1–3]. In addition, coronavirus disease 2019 (COVID-19) has necessitated the use of the online platform in medical education [4–8]. Globally, online clinical clerkship was proposed to replace face-to-face clinical clerkship during the COVID-19 pandemic (mid-COVID-19) [9]. Various opinions and practical innovations have been reported in relation to making online clinical clerkship a viable alternative to face-to-face clerkship [10–12]. However, there are currently no scores that can predict the effectiveness of online clinical clerkship curricula. Developing such predictive scores would make online clinical clerkship more effective.

We aimed to develop and externally validate a prognostic score that represents students' satisfaction and motivation, and can predict the effectiveness of the online clinical clerkship curriculum.

## Material and methods

### Survey design and data collection

This study followed the principles of the Declaration of Helsinki and was approved by Kyushu University's ethics committee. To achieve our objectives, we conducted cross-sectional surveys. We developed the survey questions with a focus on content validity through an iterative process within the team, which included a medical educator. In addition, we administered the survey to a pilot group of 30 medical students. During a semi-structured interview after the survey was administered, these students were asked to provide feedback on the clarity of formulation of the items and on the format of the questionnaire.

The study consisted of two parts. First, we collected the survey responses for the derivation dataset. Through derivation analysis, we developed a scoring model for the online curriculum to predict medical students' satisfaction and motivation. We then collected the survey responses for the validation dataset. Developed using the derivation dataset, we assessed our predictive model's prediction performance via validation analysis.

The study participants were undergraduate medical school students in Japan. The survey that yielded the derivation dataset was conducted from May 29, 2020 to June 14, 2020. This period was chosen because the first declaration of a state of emergency due to COVID-19 in Japan (April 7–May 25, 2020) led to the widespread use of online clinical clerkship. We conducted the survey that yielded the validation dataset from February 26 to March 11, 2021. This period was chosen because the second declaration of a state of emergency due to COVID-19 in Japan (January 7–March 21, 2021) also led to the widespread use of online clinical clerkship.

Both surveys have identical content. The questionnaire was presented along with an informative letter summarizing the research purpose, as well as the informed consent form, written briefly and clearly in Japanese to avoid misinterpretation. The questionnaire took 3–5 min to complete.

The inclusion criteria were as follows: students (1) who switched from face-to-face to online clinical clerkship mid-COVID-19 and (2) completed all the survey questions. We included fifth- and sixth-year medical students in the derivation dataset and fourth- and fifth-year

students in the validation study. This is because in Japan, medical students graduate in March, so sixth-year students often do not participate in clinical clerkships between January and March. The Japanese undergraduate medical education curriculum is six years period. It usually consists of four/three and half years of preclinical education and two/two and half years of clinical clerkship. Clinical clerkship has medical students participate as members of a medical team in a variety of clinical settings. Medical students can acquire clinical competencies through clinical clerkship.

## Survey items

The questionnaire asked respondents to share information about their demographic information. If participants were medical students who switched from face-to-face to online clinical clerkship, we inquired as to the effectiveness of online compared to face-to-face clinical clerkship and, how frequently they had various experiences while completing their online clinical clerkship.

## Exposure factors and other factors

We examined the extent of students' exposure to factors associated with the effectiveness of online clinical clerkship. We decided which factors should be included in our survey based on previous literature and recommendations for medical education. The factors incorporated into this study included lecture duration [13–15] and lecture frequency per week [13, 16, 17]; opportunities to take quizzes [18, 19], submit assignments [20, 21], give oral presentations [22, 23], observe physicians' practice [10, 24], practice clinical skills [10, 24], participate in interprofessional meetings [24–26], and interact with physicians [24, 27]; and the frequency of technical Internet-related problems [28]. For details regarding the options for each answer, see **S1 Method**

## Measuring outcomes

Based on Kirkpatrick's assessment model for assessing the usefulness of medical education and medical curricula [26, 29–31], we used two outcome measures of the effectiveness of online clinical clerkship: satisfaction level (Level 1 in Kirkpatrick's assessment model) and motivation level (Level 2a in Kirkpatrick's assessment model). In questions about satisfaction and motivation, the answers were based on a 5-point Likert scale, ranging from 1 = *the level/ amount was much higher during face-to-face clerkship than it was during online clinical clerkship* and 5 = *the level was much higher during online clinical clerkship than it was during face-to-face clerkship*.

## Total number of respondents and the number of respondents included in the study (**Fig 1**)

For the derivation dataset, we collected responses from 2,640 Japanese medical students. Among the 2,640 respondents, 2,594 (98.3%) consented to participate in this study. The sample size was calculated based on a 99% confidence interval (CI) and a 5% margin of error. The student population enrolled in their fifth and sixth years at Japanese medical universities totaled 18,195. The required sample size was 642, and in this study, the total number of respondents was 2,594, which is 4-fold larger than required. Respondents with missing answers were excluded from the study. Of the remaining respondents, we included those who indicated having experienced a face-to-face clerkship in a hospital prior to the first emergency declaration

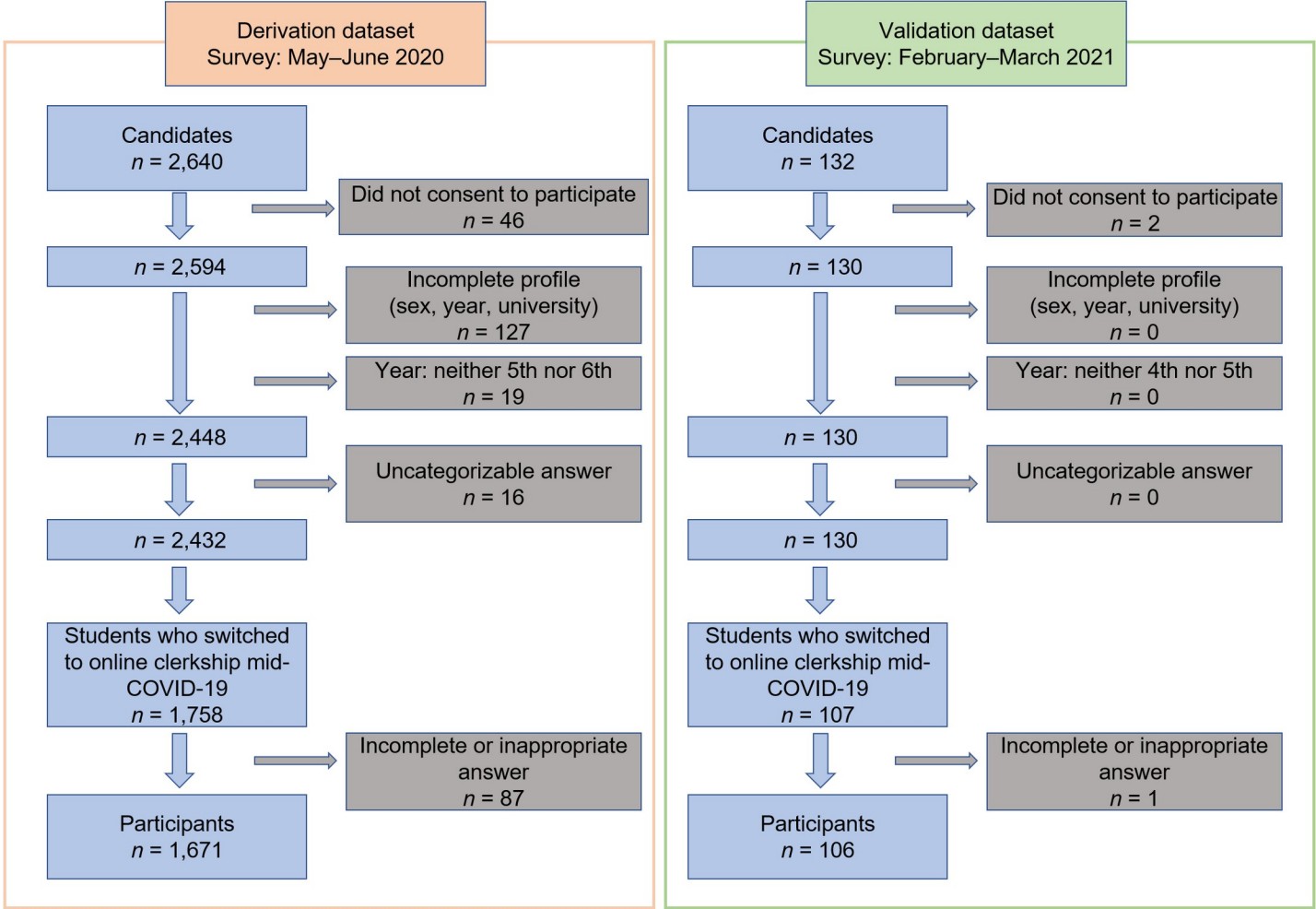

**Fig 1. A flowchart showing the selection of suitable participants for inclusion in this study.** To compile the derivation dataset, we collected responses from 2,640 Japanese medical students. Among the 2,640 respondents, 2,594 (98.3%) consented to participate in this study. Respondents with answers missing from their profile were excluded from the analysis ($n = 127$). In Japan, fifth- and sixth-year students are the equivalent years of practice, so we also excluded responses from students in years below the fifth year ($n = 19$). The next step was to categorize the responses according to the type of education the students received mid-COVID-19. Sixteen respondents were excluded for inappropriate responses. Of the 2,432 remaining respondents, 1,758 reported that they had experienced a face-to-face clerkship in a hospital pre-COVID-19 and then switched to an online clinical clerkship due to the COVID-19 crisis. Of the 1,758 respondents, those with missing responses regarding the content of the education to which they were exposed were excluded ($n = 87$). Finally, 1,671 responses were used for derivation analysis in this study.

and then subsequently switching to an online clinical clerkship. Finally, 1,671 responses were used in the derivation analysis (**Fig 1**).

For the validation dataset, we collected responses from 132 Japanese medical students. Among the 132 respondents, 130 (98.5%) consented to participate in this study. Of the remaining respondents, we included those who indicated having experienced a face-to-face clerkship in a hospital prior to the second emergency declaration and then subsequently switching to an online clinical clerkship. Finally, 106 responses were used for the validation analysis (**Fig 1**).

To compile the validation dataset, we collected responses from 132 Japanese medical students. Among the 132 respondents, 130 (98.5%) consented to participate in this study. Respondents with answers missing from their profile were excluded from the analysis ($n = 0$). In Japan, fourth- and fifth-year students are the equivalent years of practice, so we also excluded responses from students in other years ($n = 0$). The next step was to categorize the responses

according to the type of education the students received mid-COVID-19. No respondents were excluded due to an inappropriate response. Of the 130 remaining respondents, 107 reported that they had experienced a face-to-face clerkship in a hospital pre-COVID-19 and then switched to an online clinical clerkship due to the COVID-19 crisis. Of the 107 respondents, those with missing responses regarding the content of the education to which they were exposed were excluded ($n = 1$). Finally, 106 responses were used for validation analysis in this study.

## Statistical analysis

We used SPSS v27 (IBM, Armonk, NY, USA). Statistical significance was set at a two-sided $p$-value of 0.05.

Using the derivation dataset, we performed a multivariate logistic regression analysis to identify the factors associated with each outcome measurement of the effectiveness of online clinical clerkship, per the outcomes defined in the Measuring Outcomes section. We categorized the outcomes into the following binary: equivalent/better than face-to-face clerkship or not. The predictive variables included (a) lecture duration, (b) lecture frequency, (c) quizzes, (d) assignment submission, (e) student oral presentations, (f) observation of practice, (g) clinical skills practice, (h) participation in interprofessional meetings, (i) interactive discussions with physicians, and (j) technical Internet-related problems. We categorized the answers to (a)–(h) into three categorical variables: more opportunities in face-to-face clerkship, same opportunities in online as in face-to-face clinical clerkship, and more opportunities in online clinical clerkship. We categorized (j) (technical Internet-related problems) into three variables: not at all, a little, and a lot. In this study, we did not incorporate variables related to medical students' profiles because doing so would not have been in keeping with the study's aim to develop a prediction model for the effectiveness of the online clinical clerkship curriculum.

Based on multivariate logistic regression analysis, we only incorporated associated factors ($p < 0.1$) into the final prediction model. Using multivariate logistic regression analysis, the final model determined point values for each associated variable and developed an integer-based estimation system for satisfaction and motivation, respectively.

By applying this model to the validation dataset, we assessed the prognostic instrument's predictive accuracy, with discrimination. Discrimination (i.e., the degree to which a model differentiates between participants answering equivalent/better satisfaction/motivation in face-to-face clerkship) was calculated using concordance (c-) statistics, ranging from 0.5 (no discrimination) to 1.0 (perfect discrimination).

## Results

Table 1 presents the participants' demographic information and the proportion of respondents that selected each answer to every survey question.

### Identifying the factors associated with satisfaction/motivation level regarding online clinical clerkship

S1 Table shows the results of the multivariate logistic regression analysis for students' satisfaction with online clinical clerkship. The positive factors associated with satisfaction were lecture frequency (estimate = 0.054; $p = .010$), quizzes (estimate = 0.291; $p = .001$), student oral presentations (estimate = 0.175; $p = .046$), observation of practice (estimate = 0.584; $p = .001$), clinical skills practice (estimate = 0.594; $p = .002$), participation in interprofessional meetings (estimate = 0.421; $p = .003$), and interactive discussion with physicians (estimate = 0.494; $p < .001$). The negative associated factor was technical problems (estimate = -0.155, $p = .043$).

**Table 1. Summary of respondents' answers in each survey.**

| Participants | Derivation dataset (*n* = 1,671) | Validation dataset (*n* = 106) |
|---|---|---|
| **Participants' profiles** | | |
| Sex, No. (%) | Male: 979 (58.6), Female: 692 (41.4) | Male: 60 (56.6), Female: 46 (43.4) |
| Year in medical school, No. (%) | 5th: 707 (42.3), 6th: 964 (57.7) | 4th: 51 (48.1), 5th: 55 (51.9) |
| **Factors in online clinical clerkship** | | |
| Lecture duration[a] No. (%) | 1: 287 (17.2), 2: 54 (3.2), 3: 139 (8.3), 4: 250 (15.0), 5: 406 (24.3), 6: 329 (19.7), 7: 118 (7.1), 8: 88 (5.3) | 1: 10 (9.4), 2: 2 (1.9), 3: 4 (3.8), 4: 21 (19.8), 5: 37 (34.9), 6: 26 (24.5), 7: 2 (1.9), 8: 4 (3.8) |
| Lecture frequency per week[b] No. (%) | 1: 287 (17.2), 2: 141 (8.4), 3: 166 (9.9), 4: 138 (8.3), 5: 181 (10.8), 6: 139 (8.3), 7: 357 (21.4), 8: 61 (3.7), 9: 50 (3.0), 10: 28 (1.7), 11: 12 (0.7), 12: 63 (3.8), 13: 48 (2.9) | 1: 10 (9.4), 2: 4 (3.8), 3: 16 (15.1), 4: 21 (19.8), 5: 24 (22.6), 6: 9 (8.5), 7: 16 (15.1), 8: 2 (1.9), 9: 1 (0.9), 10: 1 (0.9), 11: 0 (0), 12: 2 (1.9), 13: 0 (0) |
| Opportunity to take quizzes[c] No. (%) | 1: 357 (21.4), 2: 240 (14.4), 3: 753 (45.1), 4: 206 (12.3), 5: 115 (6.9) | 1: 16 (15.1), 2: 17 (16.0), 3: 62 (58.5), 4: 10 (9.4), 5: 1 (0.9) |
| Opportunity to submit assignments[c] No. (%) | 1: 212 (12.7), 2: 171 (10.2), 3: 473 (28.3), 4: 382 (22.9), 5: 433 (25.9) | 1: 6 (5.7), 2: 6 (5.7), 3: 35 (33.0), 4: 36 (34.0), 5: 23 (21.7) |
| Opportunity to give oral presentations[c] No. (%) | 1: 700 (41.9), 2: 370 (22.1), 3: 400 (23.9), 4: 119 (7.1), 5: 82 (4.9) | 1: 35 (33.0), 2: 33 (31.1), 3: 27 (25.5), 4: 9 (8.5), 5: 2 (1.9) |
| Opportunity to observe physicians' practice[c] No. (%) | 1: 1,286 (77.0), 2: 238 (14.2), 3: 87 (5.2), 4: 25 (1.5), 5: 35 (2.1) | 1: 74 (69.8), 2: 22 (20.8), 3: 7 (6.6), 4: 2 (1.9), 5: 1 (0.9) |
| Opportunity to practice clinical skills[c] No. (%) | 1: 1,250 (74.8), 2: 263 (15.7), 3: 124 (7.4), 4: 14 (0.8), 5: 20 (1.2) | 1: 74 (69.8), 2: 19 (17.9), 3: 12 (11.3), 4: 0 (0), 5: 1 (0.9) |
| Opportunity to participate in interprofessional meetings[c] No. (%) | 1: 1,202 (71.9), 2: 244 (14.6), 3: 152 (9.1), 4: 37 (2.2), 5: 36 (2.2) | 1: 72 (67.9), 2: 20 (18.9), 3: 10 (9.4), 4: 3 (2.8), 5: 1 (0.9) |
| Opportunity to interact with physicians[c] No. (%) | 1: 798 (47.8), 2: 382 (22.9), 3: 303 (18.1), 4: 108 (6.5), 5: 80 (4.8) | 1: 42 (39.6), 2: 36 (34.0), 3: 23 (21.7), 4: 5 (4.7), 5: 0 (0) |
| Frequency of technical Internet-related problems[d] No. (%) | 1: 575 (34.4), 2: 416 (24.9), 3: 296 (17.7), 4: 297 (17.8), 5: 87 (5.2) | 1: 23 (21.7), 2: 27 (25.5), 3: 20 (18.9), 4: 29 (27.4), 5: 7 (6.6) |
| **Outcome measurement** | | |
| Satisfaction level[e] No. (%) | 1: 517 (30.9), 2: 570 (34.1), 3: 348 (20.8), 4: 138 (8.3), 5: 98 (5.9) | 1: 47 (44.3), 2: 35 (33.0), 3: 15 (14.2), 4: 8 (7.5), 5: 1 (0.9) |
| Motivation level[e] No. (%) | 1: 543 (32.5), 2: 525 (31.4), 3: 381 (22.8), 4: 108 (6.5), 5: 114 (6.8) | 1: 51 (48.1), 2: 38 (35.8), 3: 12 (11.3), 4: 4 (3.8), 5: 1 (0.9) |

a: Eight choices, arranged in 15-min increments: 1) not at all, 2) ≤15 min per lecture, 3) ≥15 min to ≤30 min per lecture, 4) ≥30 min to ≤45 min per lecture, 5) ≥45 min to ≤60 min per lecture, 6) ≥60 min to ≤75 min per lecture, 7) ≥75 min to ≤90 min per lecture, and 8) ≥90 min per lecture.

b: 13 choices: 1) not at all, 2) less than once a week, 3) about once a week, 4) about twice a week, 5) about three times a week, 6) about four times a week, 7) about five times a week, 8) about six times a week, 9) about seven times a week, 10) about eight times a week, 11) about nine times a week, 12) about ten times a week, and 13) over ten times a week.

c: 5-point Likert scale, ranging from 1 = *there were many more opportunities during face-to-face clinical clerkship than there were during online clinical clerkship* to 5 = *there were many more opportunities during online clinical clerkship than there were during face-to-face clinical clerkship*.

d: 5-point Likert scale, ranging from 1 = *not at all* to 5 = *very frequently*.

e: 5-point Likert scale, ranging from 1 = *the level/amount was much higher during face-to-face clinical clerkship than it was during online clinical clerkship* to 5 = *the level/amount was much higher during online clinical clerkship than it was during face-to-face clinical clerkship*.

**Table 2. Scores for each variable.**

| | Satisfaction | | | | | | Motivation | | | | | |
|---|---|---|---|---|---|---|---|---|---|---|---|---|
| | Estimate | Estimate/ 0.06 | Opportunities compared to face-to-face clerkship | | | | Estimate | Estimate/ 0.06 | Opportunities compared to face-to-face clerkship | | | |
| | | | Less | Same | More | Score | | | Less | Same | More | Score |
| Lecture frequency | 0.049 | 0.82 | a | | | /12 | - | - | - | - | - | - |
| Quizzes | 0.321 | 5.35 | 0 | 5 | 10 | /10 | 0.270 | 4.50 | 0 | 5 | 10 | /10 |
| Oral presentations | 0.192 | 3.20 | 0 | 3 | 6 | /6 | 0.331 | 5.52 | 0 | 6 | 12 | /12 |
| Observation | 0.583 | 9.72 | 0 | 10 | 20 | /20 | 0.422 | 7.03 | 0 | 7 | 14 | /14 |
| Practice | 0.586 | 9.77 | 0 | 10 | 20 | /20 | 0.651 | 10.85 | 0 | 11 | 22 | /22 |
| Interprofessional meetings | 0.416 | 6.93 | 0 | 7 | 14 | /14 | 0.497 | 8.28 | 0 | 8 | 16 | /16 |
| Interactive discussion | 0.492 | 8.20 | 0 | 8 | 16 | /16 | 0.358 | 5.97 | 0 | 6 | 12 | /12 |
| Technical problems | -0.158 | -2.63 | 0 (Not at all) | -3 (A little) | -6 (A lot) | /-6 | -0.248 | -3.97 | 0 (Not at all) | -4 (A little) | -8 (A lot) | /-8 |
| Sum | | | | | | /98 | | | | | | /86 |
| Cut-off value | | | | | | 22 | | | | | | 17 |

a: Not at all: 0, less than once a week: 1, once a week: 2, twice a week: 3, three times a week: 4, four times a week: 5, five times a week: 6, six times a week: 7, seven times a week: 8, eight times a week: 9, nine times a week: 10, ten times a week: 11, over ten times a week: 12.

S2 Table shows the results of the multivariate logistic regression analysis for students' motivation with regard to online clinical clerkship. The positive factors associated with motivation were quizzes (estimate = 0.264; $p$ = .002), student oral presentations (estimate = 0.313; $p <$ .001), observation of practice (estimate = 0.427; $p$ = .010), clinical skills practice (estimate = 0.657; $p$ = .001), participation in interprofessional meetings (estimate = 0.497; $p$ = .001), and interactive discussion with physicians (estimate = 0.338; $p <$ .001). The negative associated factor was technical Internet-related problems (estimate = -0.270; $p <$ .001).

## Final prediction model

Having identified the associated variables via multivariate logistic regression analysis, we again performed multivariate logistic regression analysis, this time incorporating only the associated variables identified in the first analysis. In the prediction model for satisfaction, eight predictors remained in the final multivariable model after simplification: lecture frequency (estimate = 0.049; $p$ = .007), quizzes (estimate = 0.321; $p <$ .001), oral presentations (estimate = 0.192; $p$ = .027), observation (estimate = 0.583; $p$ = .001), practice (estimate = 0.586; $p$ = .002), interprofessional meetings (estimate = 0.416; $p$ = .004), interactive discussion (estimate = 0.492; $p <$ .001), and technical problems (estimate = -0.158; $p$ = .038) (S3 Table). In the prediction model for motivation, seven predictors remained in the final multivariable model after simplification: quizzes (estimate = 0.270; $p$ = .001), oral presentations (estimate = 0.331; $p <$ .001), observation (estimate = 0.422; $p$ = .010), practice (estimate = 0.651; $p$ = .001), interprofessional meetings (estimate = 0.497; $p$ = .001)), interactive discussion (estimate = 0.358; p < .001), and technical problems (estimate = -0.248; $p$ = .001) (S4 Table). Based on the results of this final multivariate logistic regression analysis, we assigned point values to these variables and developed an integer-based estimation system for satisfaction and motivation, respectively (Table 2).

## Validation analysis to assess the scoring model's prediction performance

We determined our predictive instrument's discrimination by applying it to the validation dataset. Based on these scores, we created receiver operating characteristic (ROC) curves to

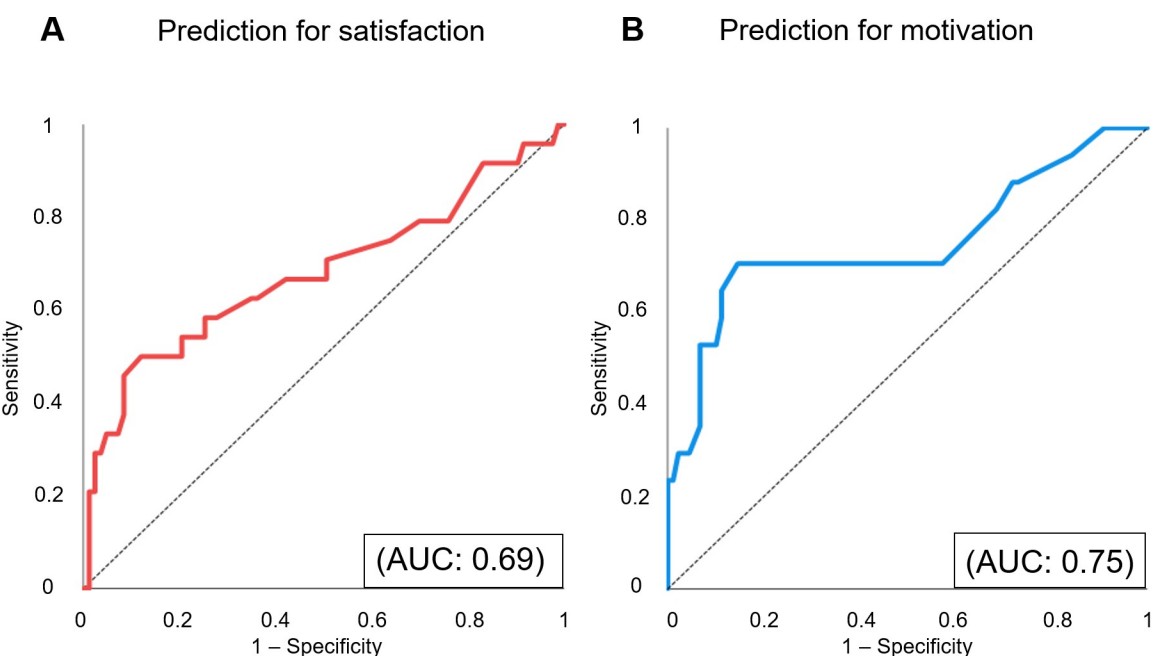

**Fig 2. The prediction models' accuracy for satisfaction and motivation.** We applied the scoring prediction model developed using the derivation dataset to the validation dataset. Based on the model's prediction performance in the validation dataset, we created receiver operating characteristic (ROC) curves for satisfaction and motivation. (A) A given ROC curve delineates predictive accuracy for satisfaction. (B) A given ROC curve delineates predictive accuracy for motivation. AUC: area under the curve.

predict whether students' satisfaction/motivation was equivalent/better than in face-to-face clerkship, as shown in **Fig 2.**

The c-statistics, which indicate the area under the curve to predict satisfaction and motivation, were $0.69 \cdot$ (95% CI: 0.55–0.83, $p = .005$) and $0.75 \cdot$ (95% CI: 0.60–0.90, $p = .001$), respectively. When the cut-off value for the satisfaction scores was set to 22, sensitivity was 0.50, and specificity was 0.89. When the cut-off value for the motivation scores was set to 17, sensitivity was 0.71 and specificity was 0.85.

## Discussion

We developed a practical instrument to predict the effectiveness of online clinical clerkship based on medical students' satisfaction and motivation. Although an instrument to predict the online clinical clerkship curriculum's effectiveness has not yet been developed, such an instrument is needed, especially given the increase in demand for online clinical clerkship due to the COVID-19 pandemic.

As the results show, our model's prediction performance was relatively high (satisfaction: 0.69, motivation: 0.75). Satisfaction and motivation are based on individual self-assessment, which means that different people can rate their satisfaction with and motivation for the same curriculum differently. Individual factors (such as sex and year in medical school) are reported to be related to satisfaction and motivation [30, 32]. Nevertheless, we did not incorporate individual components into the score in our prediction model. This was because our goal was to develop a scoring model that predicts the effectiveness of online clinical clerkship *curricula*. In short, our model's prediction performance was high in spite of not incorporating individual demographic information.

Using our scoring system, educators will be able to both predict the effectiveness of and improve their online clinical clerkships. Our instrument will be of great help to educators in building a better online clinical clerkship curriculum. In our study, the best cutoff values for motivation and satisfaction were 22 and 17, respectively (**Table 2**). Educators can use these scores as guides to improve online clinical clerkship by enhancing the educational approaches in their curriculum, with the goal of exceeding the scores we obtained. For example, to maintain medical students' satisfaction and motivation during online clinical clerkships in a manner that is comparable to face-to-face clerkship, educators might consider providing the students with the same opportunities to observe physicians' practice and to practice their clinical skills as face-to-face clinical clerkship (at only these two variables, score for satisfaction:20, and score for motivation:18) (**Table 2**).

This study has several strengths. The assembled population is one of the largest study sample sizes for predicting the effectiveness of online clinical clerkship. Our instruments were developed using big data from national-level cross-sectional studies involving a total of 1,671 medical students in the derivation dataset and 106 students in the validation dataset. Another strength is that our scoring system is a practical instrument that can be applied to the online clinical clerkship curriculum. The predictors are well defined and easily measured, so that students' online opportunities to experience each educational approach can be evaluated as equivalent to, or more or less frequent than in face-to-face clerkship. We also made technical troubles simple to evaluate in three frequency categories: never, occasionally, and often.

Our study has some limitations. First, the cross-sectional study retrospectively measures subjects' exposure and outcome simultaneously. Thus, recall bias was unavoidable. Second, our prediction model is based on medical students' satisfaction and motivation, which are self-assessed outcomes that are easily measured using a questionnaire. However, knowledge and skill acquisition, which are Level 2b in Kirkpatrick's assessment model, are more objective outcomes [26, 29–31]. Predictive instruments based on such outcomes, as opposed to satisfaction and/or motivation, might be more useful. Therefore, more research might be required to show the correlation with exam scores or patient outcomes before using this study as a tool to modify curricula. The third limitation is that qualitative factors on the part of the faculty were not considered in the current model because the current study was a quantitative survey. For example, the content and quality of lectures, the faculty's skills of online facilitation might also affect the students' satisfaction and motivation. Forth, in this study, we only included medical students in Japan. Future research should consider the applicability of these results to other countries. Online clinical clerkship would greatly depend on health care resources, Internet infrastructure in countries/communities, and whether students are familiar with the devices required to access online education [33, 34]. This indicates that our model would be difficult to apply in countries with limited healthcare resources and/or a lack of Internet infrastructure. Another limitation is that these data are based on online clinical clerkship during the COVID-19 pandemic. The impact of COVID-19 varies across countries and communities. Furthermore, online clinical clerkship post-COVID-19 pandemic may be of a different nature (e.g., a combination of face-to-face and online). We believe that our prediction model can be applied in other countries' online clinical clerkship curriculum or post-COVID-19 pandemic. However, we need to evaluate our model's validity in other countries and post-COVID-19 pandemic in future studies.

## Conclusion

We successfully developed a scoring model to predict the effectiveness of the online clinical clerkship curriculum, based on students' satisfaction and motivation. Our scoring model will accurately predict and improve the online clinical clerkship curriculum's effectiveness.

## Supporting information

**S1 Method. Detailed description of the answer options for each survey question.**
(DOCX)

**S1 Table. Multivariate logistic regression analysis to identify the factors associated with maintaining medical students' satisfaction with online clerkship (Level 1 in Kirkpatrick's assessment model).**
(DOCX)

**S2 Table. Multivariate logistic regression analysis to identify the factors associated with maintaining medical students' motivation during online clerkship (Level 2a in Kirkpatrick's assessment model).**
(DOCX)

**S3 Table. The final model using multivariate logistic regression analysis to predict medical students' satisfaction with online clerkship (Level 1 in Kirkpatrick's assessment model).**
(DOCX)

**S4 Table. The final model using multivariate logistic regression analysis to predict medical students' motivation during online clerkship (Level 2a in Kirkpatrick's assessment model).**
(DOCX)

**S1 Dataset.**
(XLSX)

**S2 Dataset.**
(XLSX)

## Acknowledgments

The authors are grateful to Dr. Eishi Asano at Children's Hospital of Michigan, Detroit Medical Center, Wayne State University for the advice on statistical analysis. The authors are also grateful to all participants of this survey-based study.

## Author Contributions

**Conceptualization:** Naoto Kuroda, Anna Suzuki, Nobuhiro Nagai, Makoto Kikukawa.

**Data curation:** Naoto Kuroda, Anna Suzuki, Kai Ozawa, Nobuhiro Nagai, Yurika Okuyama, Kana Koshiishi, Masafumi Yamada.

**Formal analysis:** Naoto Kuroda.

**Funding acquisition:** Makoto Kikukawa.

**Investigation:** Naoto Kuroda.

**Methodology:** Naoto Kuroda.

**Project administration:** Naoto Kuroda.

**Resources:** Kai Ozawa, Nobuhiro Nagai, Yurika Okuyama, Kana Koshiishi, Masafumi Yamada.

**Supervision:** Makoto Kikukawa.

**Validation:** Naoto Kuroda.

**Visualization:** Naoto Kuroda.

**Writing – original draft:** Naoto Kuroda.

**Writing – review & editing:** Anna Suzuki, Makoto Kikukawa.

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
