## [Decision Letter · Decision Letter 0]

5 Jan 2022

PONE-D-21-32687Predicting the effectiveness of the online clinical clerkship curriculum: Development of a multivariate prediction model and validation studyPLOS ONE

Dear Dr. Kuroda,

Thank you for submitting your manuscript to PLOS ONE. After careful consideration, we feel that it has merit but does not fully meet PLOS ONE’s publication criteria as it currently stands. Therefore, we invite you to submit a revised version of the manuscript that addresses the points raised during the review process.

We look forward to receiving your revised manuscript.

Kind regards,

Feroze Kaliyadan, M.D.

Academic Editor

PLOS ONE

Journal Requirements:

Additional Editor Comments:

The concept of developing a predictive model for evaluating effectiveness of the online clinical clerkship curriculum is important, interesting and very relevant.

You will need to add more details on the questionnaire validation process.

A brief summary of the curricular model followed (especially for year 5 and 6 )normally needs to be added to put things in context

Limitations need to be elaborated, would have been especially good if faculty perspective was also incorporated

Finally, the discussion needs to elaborate on how far the results can be extrapolated to other regions

Reviewers' comments:

Reviewer's Responses to Questions

**Comments to the Author**

1. Is the manuscript technically sound, and do the data support the conclusions?

Reviewer #1: Yes

Reviewer #2: Yes

2. Has the statistical analysis been performed appropriately and rigorously? 

Reviewer #1: Yes

Reviewer #2: I Don't Know

3. Have the authors made all data underlying the findings in their manuscript fully available?

Reviewer #1: Yes

Reviewer #2: Yes

4. Is the manuscript presented in an intelligible fashion and written in standard English?

Reviewer #1: Yes

Reviewer #2: Yes

5. Review Comments to the Author

Reviewer #1: Article is recommended for submission since it meets all the criteria of the journal. The tool for validation of online clerkship is something that is a requisite today. The statistical works and the language of the paper are acceptable. The reference styling is acceptable.

Reviewer #2: The study is very relevant considering the pandemic and could be very useful to Medical Colleges all over the world to evaluate and improve on their curricula.

A potential source of the results not reflecting actual improvement in the curricula, is the fact, that in the current scenario online learning is more of a necessity rather than an option. The student satisfaction being high could be reflecting their greater comfort rather than better learning outcomes as the study is only assessing student satisfaction and has no information regarding correlation with exam scores or patient outcomes. More research might be required in these avenues before using this study as a tool to modify curricula.

6. PLOS authors have the option to publish the peer review history of their article (what does this mean?). If published, this will include your full peer review and any attached files.

Reviewer #1: No

Reviewer #2: No

---

## [Author Response · Author response to Decision Letter 0]

12 Jan 2022

Reviewer #1: Article is recommended for submission since it meets all the criteria of the journal. The tool for validation of online clerkship is something that is a requisite today. The statistical works and the language of the paper are acceptable. The reference styling is acceptable.

→Thank you for your favorable comments.

Reviewer #2: The study is very relevant considering the pandemic and could be very useful to Medical Colleges all over the world to evaluate and improve on their curricula.

A potential source of the results not reflecting actual improvement in the curricula, is the fact, that in the current scenario online learning is more of a necessity rather than an option. The student satisfaction being high could be reflecting their greater comfort rather than better learning outcomes as the study is only assessing student satisfaction and has no information regarding correlation with exam scores or patient outcomes. More research might be required in these avenues before using this study as a tool to modify curricula.

→Thank you for your comments. As you pointed out, the clear limitation of this study is that we did not investigate the correlation with exam scores or patient’s outcomes. We added this limitation as follows:

 Therefore, more research might be required to show the correlation with exam scores or patient outcomes before using this study as a tool to modify curricula.

---

## [Editor Report · Decision Letter 1]

14 Jan 2022

Predicting the effectiveness of the online clinical clerkship curriculum: Development of a multivariate prediction model and validation study

PONE-D-21-32687R1

Dear Dr. Kuroda,

We’re pleased to inform you that your manuscript has been judged scientifically suitable for publication and will be formally accepted for publication once it meets all outstanding technical requirements.

Kind regards,

Feroze Kaliyadan, M.D.

Academic Editor

PLOS ONE

---

## [Editor Report · Acceptance letter]

19 Jan 2022

PONE-D-21-32687R1 

Predicting the effectiveness of the online clinical clerkship curriculum: Development of a multivariate prediction model and validation study 

Dear Dr. Kuroda:

I'm pleased to inform you that your manuscript has been deemed suitable for publication in PLOS ONE. Congratulations! Your manuscript is now with our production department. 

Kind regards, 

on behalf of

Dr. Feroze Kaliyadan 

Academic Editor

PLOS ONE